# COVID-19 Modeling Outcome versus Reality in Sweden

**DOI:** 10.3390/v14081840

**Published:** 2022-08-22

**Authors:** Marcus Carlsson, Cecilia Söderberg-Nauclér

**Affiliations:** 1Centre for Mathematical Sciences, Lund University, 22100 Lund, Sweden; 2Department of Medicine, Solna, BioClinicum, Karolinska Institutet, 171 64 Solna, Sweden; 3Department of Neurology, Karolinska University Hospital, 171 77 Stockholm, Sweden; 4Department of Biosciences, InFLAMES Research Flagship Center, MediCity, University of Turku, 20500 Turku, Finland

**Keywords:** modeling, SARS-CoV-2, COVID-19, SIR, SEIR

## Abstract

It has been very difficult to predict the development of the COVID-19 pandemic based on mathematical models for the spread of infectious diseases, and due to major non-pharmacological interventions (NPIs), it is still unclear to what extent the models would have fit reality in a “do nothing” scenario. To shed light on this question, the case of Sweden during the time frame from autumn 2020 to spring 2021 is particularly interesting, since the NPIs were relatively minor and only marginally updated. We found that state of the art models are significantly overestimating the spread, unless we assume that social interactions significantly decrease continuously throughout the time frame, in a way that does not correlate well with Google-mobility data nor updates to the NPIs or public holidays. This leads to the question of whether modern SEIR-type mathematical models are unsuitable for modeling the spread of SARS-CoV-2 in the human population, or whether some particular feature of SARS-CoV-2 dampened the spread. We show that, by assuming a certain level of pre-immunity to SARS-CoV-2, we obtain an almost perfect data-fit, and discuss what factors could cause pre-immunity in the mathematical models. In this scenario, a form of herd-immunity under the given restrictions was reached twice (first against the Wuhan-strain and then against the alpha-strain), and the ultimate decline in cases was due to depletion of susceptibles rather than the vaccination campaign.

## 1. Introduction

### 1.1. Background

In March 2020, when it became clear that the COVID-19 outbreak was turning into a global pandemic, mathematical models were used to predict the magnitude of viral spread and health care needs, which had a major impact on public policy. For example, the Imperial College report No. 9 predicted an 81% hit-rate within a couple of months, in the worst case scenario [1], if the government did nothing to mitigate or suppress the pandemic. This led to a subsequent strategy shift towards suppression and eventually lock-down in the UK on 23 March 2020, as well as in many other countries world-wide. In sharp contrast, the Swedish government decided to keep the society relatively open and maintained relatively minor and rather fixed non-pharmacological interventions (NPIs) until the summer of 2021, when the majority of the population had been vaccinated. Sweden therefore provides an excellent example that can be used to test the validity of epidemiological mathematical models. Rather surprisingly, in the light of typical modeling outcomes, it is estimated that the total hit-rate in major Swedish cities was just above 20% in March 2021, one year after the onset of the pandemic [2] (Appendix A explains how to access the data in [2]). In this article, we analyze potential explanations for this drastic reduction.

In April 2020, when seroprevalence measures indicated a hit rate of around 5% [2], we observed a decline in admissions to intensive care units (ICUs) that continued through May and led to very low new ICU-admissions during the summer of 2020, indicating that the viral spread in society was very low as well. This is also visible in Figure 1, displaying daily cases of COVID-19 in Stockholm County. Note that the data from the “first wave” is very unreliable since there was no large-scale public testing before May 2020.

When a major deadly epidemic hits, the society reacts in a way that is impossible to predict mathematically, sometimes referred to as “herd-protection”. Therefore, mathematical models are often grossly wrong in their predictions, as exemplified by previous potential disasters, such as various Ebola outbreaks [3,4]. This phenomenon, i.e., major voluntary reductions in social interactions, could, in theory, explain the difference between model outcome and reality also for SARS-CoV-2 in Sweden. However, the decline of the first wave happened despite the fact that schools were open and face-masks were not used. Even so, the seroprevalence among children aged 0–19 was very low, just above 6%, in June 2020 after the first wave [2]. It seemed difficult to envision such a development was the result of NPIs and voluntary measures alone, indicating that a part of the population could have a higher protection against contracting SARS-CoV-2, i.e., having some sort of pre-immunity.

### 1.2. Contributions

We use publicly available data from the Swedish Public Health Agency regarding incidence, variants, vaccination coverage and seroprevalence of SARS-CoV-2 in Stockholm County, Sweden, along with tailor-made mathematical models, in order to test these different explanations of what kept the spread at bay in Sweden. We base our model on an extension of a SEIR-model with age-stratification, developed by Britton et al. [5], which we update to include antibody waning and vaccination roll-out. For comparison with real data we chose the metropolitan area of Stockholm with about 2,400,000 inhabitants, since effects due to geographical isolation are likely negligible, compared with more scarcely populated areas elsewhere. We test the hypotheses: the observed fluctuations in cases are due to:(1)Variations in NPIs and/or population behavior over time;(2)Some form of protective pre-immunity and natural depletion of susceptibles.

Although the reality certainly could be a mixture of both, we find that the former hypothesis is unlikely to be the sole explanation, whereas we obtain a good data fit in the second hypothesis (which assumes that social contact patterns were constant during the time frame studied). In this scenario, our modeling implies that Sweden has reached herd immunity twice (given the mild restrictions), once to the Wuhan-strain in early December 2020, and thereafter to the alpha-strain in the beginning of April of 2021. Additionally, the effect of the vaccination scheme in the decline of the alpha-wave seen in May 2021 is, given this hypothesis, marginal.

In a separate study [6], we demonstrate that, from a modeling perspective, a number of population heterogeneities, such as variable susceptibility, variable activity patterns, as well as self-isolatory measures, seem to have an effect identical to assuming that a certain proportion of the population had a sterilizing pre-immunity. This “pre-immunity”, if it indeed existed, is thus likely a manifestation of a much more complex phenomenon than the name suggests, and to underline this we will refer to it as “artificial pre-immunity”. In other words, it is possible that including “artificial pre-immunity” in mathematical models is crucial for good model fit, even if sterilizing immunity against the virus, a priori to the disease, does not exist on an individual level. A more rigorous attempt at explaining what this artificial pre-immunity derives from is found in [7], which we also discuss to some extent in Section 5.3.

## 2. The COVID-19 Pandemic in Sweden

### 2.1. Choice of Time Frame

We removed the “first wave” from our study, for a number of reasons. First, testing was very unreliable, mainly limited to hospitalized patients only, and, hence, we have no data to compare the models with. Secondly, it is clear that major oscillations in social interactions took place during this period, and this can not be built into mathematical modeling in a reliable manner. Additionally, social interactions significantly drop during the Swedish summer vacations, which comes close to a soft lockdown. In September of 2020, when schools had reopened, society had largely returned to a “new normal”. At the time, most people were convinced there would be no more major waves in Sweden, and the chief epidemiologist continuously downplayed the situation, for example describing a 35% increase in new cases in Sweden as “a small cautious increase” [8]. During the autumn, the number of cases were never alarmingly high and the hospitals were coping with the stream of patients, and hence there was never a sense of panic. Indeed, at the peak of the second wave, seen as the first wave in Figure 2, the recommendations for the elderly to self-isolate were lifted, thus sending a message that the pandemic was under control to the population. It is crucial to underline that mathematical models work well, in theory, also during NPIs and voluntary social reductions, as long as these are kept fairly constant during the modeling time frame, and we believe that this was the case from September 2020 to May 2021. Another factor that speaks in favor of this time frame is that testing was widely available, so data from this period are reliable, (although needs to be adjusted to account for under-reporting of cases and asymptomatic infections). Furthermore, the fact that only two major strains are involved, the original Wuhan-strain and alpha, allows us to separate the amount of cases from each respective strain, which is key for modeling.

We do not include the delta and omicron waves in our study for a number of reasons, the key reason is that the majority of the population then was vaccinated, which significantly alters the model and moreover the data on seroprevalence no longer gives any information about how many had had COVID-19. Moreover, while it is clear that antibodies from both natural infection and the vaccine protected against delta, it is not clear to what degree, and, hence, it is not possible to perform any reliable mathematical modeling. Concerning the omicron-wave, it is known that neither antibodies from the vaccine nor natural infection gave a strong protection against contracting omicron, but to what extent did it give partial protection? Without a certain answer to these questions, further mathematical modeling becomes too speculative. Of course, the fact that there was no major delta wave but a substantial omicron wave is interesting, and will be discussed further in Section 5.2

Based on the above remarks, we argue that the time frame from September 2020 to May 2021 is optimal for mathematical modeling.

### 2.2. Separating the Data

Data for new cases in Stockholm County based on PCR-testing, which has been obtained from the Swedish Public Health Agency “Folkhälsomyndigheten” (see Appendix A for details) and averaged over a 7 day window, is displayed in Figure 1 and a zoom of the relevant time frame is displayed in Figure 2 (left). The most important thing to realize in order to understand the graph in Figure 2 is that what we see is two distinct waves superimposed. Thanks to available data on variants of concern collected by the Swedish Public Health Agency [9], we know what percentage is caused by the Wuhan strain and the variants of concern, respectively. The second wave seen in the figure (i.e., the third wave in reality) is almost entirely caused by the “British variant” B.1.1.7, also known as alpha. It also contained small portions of other variants of concern, P.1, B.1.167, and B.1.351, but those caused, at most, 6% of the cases (in the month of February) and even less in the last measurement, so for simplicity we shall refer to it as the alpha-wave, whereas we shall call the first wave (in Figure 2) the Wuhan-wave. The separation in cases due to the original “Wuhan-wave” and “alpha-wave” is displayed in Figure 2 (right).

### 2.3. The Swedish NPIs

Sweden provided a unique opportunity to test mathematical models for disease progression, in particular the densely populated metropolitan area of Stockholm with about 2.4 million inhabitants, due to the virtually constant NPIs. The only major change in these during the time frame Sept 2020 to May 2021 was the closing of high-schools (age-group 15–18 years) on 7 December 2020 and subsequent reopening on 1 April 2021. Bars and restaurants have been open but only with table-service and a maximum of 4 people at each table. In addition, a number of minor changes in the recommendations were made, which are collected in Table 1, (also marked in the timeline in Figure 2). It should be noted that public compliance with these restrictions was not ideal, for example less than one out of three was reported to follow the recommendation concerning facemasks in public transport, based on random testing performed by the Swedish Public Broadcasting company [10].

Neither these policy-updates nor any of the major holidays (Autumn, Christmas, Sport, and Easter school holidays), see Figure 2 (left), seems to have had any notable effect on the two waves. In particular, the Wuhan-wave was declining rapidly *before* the Christmas-break and subsequent mask-recommendations, and the alpha-strain had leveled out before the Easter break. It is also noteworthy that the *relaxations* 1–2 coincide with the time when the second wave looses momentum and levels out. Altogether this is an indication that neither holidays nor updates to the NPIs had a major impact on case numbers. Moreover, one has to bear in mind that the Swedish population had been informed by the Swedish Public Health Agency that frequent washing of hands and keeping 2 m distance between people are the key factors to limit the spread of the virus in society. Aerosol transmission was consistently denied by the Swedish Public Health Agency who maintained that droplet transmission was the main source of infections, which, for example, led to good ventilation not being part of the pandemic preventive measures in Sweden. Consequently, among those who did try to protect themselves, face-shields were about as common as face-masks, due to the wide-spread misconception that SARS-CoV-2 primarily spreads by infected individuals coughing and sneezing, which has been repeated over and over by spokesmen for the Swedish Public Health Agency. Since COVID-19 can spread by aerosols [11], particularly in indoor climate, we argue that the conditions for steady spread of COVID-19 were good throughout the time-period studied, in particular the cold and rainy months between October 2020 and May 2021 (April and May 2021 were unusually cold).

Furthermore, the first COVID-19 vaccinations started in Sweden after Christmas 2020, but people younger than 70+ were not offered vaccinations until early April 2021 (with the exception of special work groups, such as health care workers). As the elderly are not main drivers of the epidemic, and protective antibody levels take several weeks to develop, vaccinations most likely had limited effect on case numbers during the time frame in question. (Given the second hypothesis, we prove this last claim in Section 4.2.)

### 2.4. Under-Reporting and Seroprevalence Levels

Data on daily cases is difficult to use since there is always an unknown proportion of cases that do not get tested. To account for this, we have corrected the magnitude so that it fits with data from serological testing, which is a more stable indicator of total amount of infections, since the majority who have had COVID-19 go on to develop antibodies [12]. According to the Swedish Public Health Agency [2] (see also Appendix A), in early June 2020 the seroprevalence of antibodies against SARS-CoV-2 was 11.2%, which dropped to 9.8% mid-October 2020. Since this period was characterized by a very low prevalence of COVID-19, we assume that the drop is due to antibody waning. Interpolating between these data points, we estimate the seroprevalence at 10% in early September, the beginning of our time-line. The seroprevalence then rose to 22.8% in early March 2021 [2], at the time of the onset of the alpha-wave. Based on this, we estimate that there are 2.4 actual cases of COVID-19 for every reported case. The graph in Figure 2 has been amplified accordingly.

In the serological measurements performed by the Swedish Public Health Agency it is not possible to separate between antibodies caused by the virus and the vaccine, so the measurement from March 2021 is the last that gives an indication of how many had had COVID-19 at the given time. However, children under the age of 12 were not vaccinated in Sweden, and in Table 5 of [2] data from this age-group are displayed separately, indicating that the seroprevalence rose to just below 30% after the alpha wave, and remained at that level throughout the delta surge, which was very mild in Sweden. The seroprevalence then abruptly jumps to above 70% in March 2022, following the omicron-wave.

## 3. Methods

In this section, we briefly describe key ingredients in our modeling and processing of the data, as well as the underlying assumptions. A fuller description is found in Appendix A.

### 3.1. Choice of Model

We build our model upon the SEIR-model with age-stratification which was developed by Britton et al. [5]. This is a rather standard compartmental model, with 6 age-compartments for each category, Susceptible, Exposed, Infective, and Recovered, making 24 compartments in total. The flow of people between these compartments is governed by a system of ordinary differential equations, which is the standard way of building mathematical epidemiological models, see, e.g., [13] or [14].

We include age-stratification since it is one heterogeneity for which one can find realistic parameter values, and it allows us include the vaccination roll-out (which started among the elderly and gradually was offered to younger age-groups, see Section A.3 for details). For contact intensities between various age-groups we rely on the pre-pandemic study [15] but we updated the numbers to reflect Swedish demographics, along with a relatively higher reduction in contacts among the elderly, who were encouraged to self-isolate (see Section A.2 for details). Finally, to incorporate antibody waning we also have a flow of people moving from the recovered group to the susceptible group. We assume that antibodies for either the alpha or Wuhan strain gave protection against reinfection with a half-life of 16 months, (following [16] and also less precise observations in [17]). We implemented the code in MatLab, which can be downloaded from https://github.com/Marcus-Carlsson/Covid-modeling (accessed on 10 August 2022), and the details are further described in Appendix A.

In Figure 3, we see a typical model outcome (red curve), where the parameters have been chosen to match the time series during the first two months September–October 2020. As is clear to see, it fits the data very well in those months, but in November the model curve continues to rise drastically. If this model had been used for policy decisions, then clearly a lock-down would have been called for, but no lock-down was ever imposed and some restrictions were even lifted, as explained earlier. Despite this, the measured case numbers start to level out and then sharply decay, and there is no indication of a resurge of cases before the introduction of the more contagious alpha-variant. The key research question of this paper is what caused this decay and why the model data is off (by a factor of around 3 compared to measured data). Are SEIR-type models simply poor for pandemic predictions, or is there something about human behavior or the virus itself which does not correspond to how the model is built?

We stress that nothing in the model we use is accelerating the spread compared with other models (in fact, Britton et al. developed it in an attempt to modify standard SEIR in order to *dampen* the spread). They also made a model that takes variability in the social activity level into account and demonstrated that such variations have a further damping effect on model curves. However, it is clearly impossible to measure such variations in the social activity level, and, hence, we cannot find realistic parameters for applying this model to a real society, wherefore we have built our model on the simpler age-stratified version. Nevertheless, we have also tested this model with the same parameters as in [5], and found that this does not dampen the model curves enough to alter any of the conclusions of this paper, even if the contrast between reality and model data becomes somewhat less stark, see Figure 3, (black curve).

Furthermore, the model we use is very similar to the one developed by a well renowned Swedish modeling team [18], as well as other models published for example by the Imperial College Team [19], in terms of the dynamics between the major groups Susceptible, Exposed, and Infected. These models also tend to drastically overestimate the spread. Indeed, ref. [18] predicts a cumulative number of infected people of around 30% after the first wave. The authors arrive at the “low” value of 30%, from a modeling perspective, due to assuming a 56% decrease in contacts among people of age 0–59 and a 98% reduction among those aged 60–79 (this is for scenario d), which accurately fitted ICU-occupancy and death, see Figure 2b in [18]. Given that many people in the latter age group are still working (average retirement age was 65 in Sweden 2021), an overall reduction of 98% in contacts seems somewhat unrealistic. More realistic reduction numbers, thus, give cumulative infection numbers above 30%, when the actual figure was around 10% (as we saw in Section 2.4), demonstrating the tendency of compartmental models to overestimate the spread, compared to what happened in reality (in Sweden).

Along the same lines, the famous Report 9 by the Imperial College [1] predicted a total number of 81% infected in a “do-nothing” scenario, based on a more advanced so called “agent-based model” that also treats household-contacts separately. According to Table 3 in the report [1], the number of deaths and peak ICU capacity can be reduced by 50% and 81%, respectively, in the most effective NPI-scenario, which certainly goes beyond what was implemented in Sweden. However, if these figures are directly adjusted to apply for Stockholm County, the predictions using the most severe restrictions still overestimate the actual numbers by a factor of roughly 4 (deaths) and 10 (ICU) (comparing with data from February 2021).

We remark that non-compartmental mathematical models, as found, e.g., in [20,21], have been more successful at predicting the pandemic in the short term, but we will not discuss this type of model further here.

### 3.2. Artificial Pre-Immunity and Population Heterogeneities

Before we introduce how to model with pre-immunity, we need to discuss what this actually means. Pre-immunity, as well as immunity developed after an infection, is not a binary variable that either gives a 100% protection (so-called “sterilizing immunity”) or none at all. In the separate study [6], we show that there is practically no difference mathematically between an advanced model taking variable susceptibility into account, and a more naive model which simply stipulates that a certain proportion of the population has sterilizing immunity. Hence, the idea of a binary pre-immunity works well on a macro-level, as long as one is aware that this is a simplification that does not apply on a micro-level. We remark that a number of different factors, such as genetic, cross-reactive immunity, and innate immunity, have been shown to provide variation in susceptibility [22,23,24]. In addition, whether a person gets infected or not upon exposure to SARS-CoV-2 clearly depends on the dosage as well as the time of exposure.

To complicate things further, a fraction of the population went into various degrees of self-isolation, and clearly it is impossible to distinguish such individuals from individuals with high-levels of natural defense against contracting SARS-CoV-2. However, it is shown numerically in [6] that such variations are also manifested as an artificial level of sterilizing immunity. In practice, this means, for example, that the black curve in Figure 3, produced by a model with 72 compartments (Age-Act-SEIR), could be obtained easily from the simpler 24-compartment Age-SEIR model, upon including artificial sterilizing immunity. Therefore, we have chosen to build our model on the simpler Age-SEIR-model while including a level of sterilizing immunity. This is a mathematical modeling simplification that comprises a number of complex factors, and it is impossible to know how much of this immunity comes from actual variations in susceptibility compared with variations in social interaction patterns. To underscore this, we have coined the term “Artificial Sterilizing Immunity” [6], but since this paper concerns the period when SARS-CoV-2 was relatively new, artificial sterilizing pre-immunity, or artificial pre-immunity for short, seems more accurate.

### 3.3. The Hypotheses Posed Mathematically

In order to transform the two hypotheses into mathematical language, we now describe in broad terms the key features of our model. Somewhat simplified, all mathematical models for disease progression rely on some refinement of the formula
(1)Newcases=a·β·i·s,
where *a* denotes the amount of daily contacts between individuals, β the probability that such a contact leads to disease transmission, *i* is the prevalence and *s* is the fraction of the population that are susceptible to the virus. Clearly *i* and *s* depend on time, but we shall also consider *a* and β as time dependent, to account for variations in social activity during the pandemic and the fact that the alpha strain was more contagious than the original strain.

In mathematical terms, Hypothesis 1 amounts to starting our model with s(1)=0.9 (i.e., 90% are susceptible on day 1, since 10% already had antibodies at the start of the timeline) and letting the amount of contacts vary with time. More precisely, we postulate that a(t)=a·(1−f(t)/100) where *a* is a constant chosen so that f(t)≈0 in September. In words, *f* describes the percentage-wise change in amount of contacts, comparing with September 2020 as a “normal month”.

Hypothesis 2, on the other hand, amounts to keeping the daily amount of contacts fixed, so a(t) is a constant, and instead allowing the initial value of s(1) to be substantially less than 0.9, to account for artificial pre-immunity. Since our model keeps track of infections caused by alpha and Wuhan-strain separately, we give the two strains individual β-values, as well as individual levels of artificial pre-immunity, since a natural protection against the original strain may be less effective against other variants.

## 4. Results

### 4.1. Hypothesis 1: Human Behavior

Recall that *f* describes the percentage-wise change in amount of contacts, as a function of time, comparing with September 2020 as a “normal month”. Given any fixed choice of *f*, our model produces a curve for the incidence which we can compare with measured data. We wish to underline that, in our modeling, we continuously remove vaccinated people (after the first shot) from the susceptible group to the recovered group, so *f* describes the updates in contact frequency with the vaccination roll-out already accounted for. Our model also takes into account, by default, that a majority of the elderly were self-isolating, so *f* really describes how the contact pattern in the remaining age-groups need to fluctuate. For further details we refer to Appendix A, in particular Section A.6.

In order to find an *f* so that the incidence-curve fits well with measured data, we also need to know how much more contagious the alpha-strain was compared with the original Wuhan-strain. In [25], they estimate this factor as between 38% and 130% more contagious (95% confidence interval), but this estimate comes with several uncertainties. If we suppose for the moment that it was 50% more contagious, then the red curve in Figure 4 describes how *f* (the relative change in social contacts) needs to fluctuate in order to obtain the data-fit seen by the pink graph in Figure 3. Hence, in order to match real data, we see that the contact frequency needs to continuously drop (except for a small increase in January) until it reaches levels around 50% lower in May 2021, compared with September 2020. This is clearly not realistic. On the contrary, May 2021 was characterized by a sense of relief as the case numbers were dropping, which officially was explained by the success of the vaccination campaign.

Of course, the less contagious alpha is, the less reduction in behavior is needed to keep the incidence down. This is the reason for the green shaded area, the upper boundary corresponds to assuming that alpha is only 38% more contagious, the lower by assuming it is 130% more contagious. As is plain to see, even in the 38% scenario, the fluctuations in *f* are not realistic.

To provide objective support for this affirmation, to the right in Figure 4 we see the actual behavior fluctuations according to Google Mobility Data. The data have been averaged weekly and adjusted so that the mean in September 2020 is 0. In order for the Wuhan-strain wave to turn around, as it did in November to mid-December 2020, the left image indicates that we need a reduction in *f* of around 30%. However, although Google Mobility Data does go down over the same period, it does so more modestly. For example “workplaces”, believed to be a key location for transmission, sees a reduction of around 10% in the relevant timespan, just before Christmas. Note, also, that the major fluctuations around the Christmas and Easter holidays, which are clearly visible in the Google-mobility graph, leave no clear bearing on the incidence, which is strange if human behavioral changes are the key factor for variations in incidence.

Moving to the second wave, the left image in Figure 4 indicates that we need another substantial reduction in human contacts from the month of February to May 2021, but, in stark contrast, throughout this period all relevant curves in the Google-mobility graph are going up. Let us again stress that, at this point, the seroprevalence measurements among children (as well as the data in Figure 2) indicates a total hit-rate of around 30% in June 2021 (Section 2.4), and that the incidence during autumn 2021 was kept low throughout the delta-surge (see Figure 1), even among unvaccinated children. Indeed, the September 2021 measurement in the groups 0–11, that were not given any vaccine but attended school as normal, had an estimated 28.4% seroprevalence, whereas the measurement in early December 2021 (after the delta surge) was 29.6% [2] (these data are easily accessible in Appendix A). This seems more to indicate that some sort of herd-immunity had been reached, and overall these observations cast doubt over Hypothesis 1 and indicate that other factors than human updates in mobility are needed in order to explain the two waves in Figure 2.

To further underline this point, the Google-mobility data over the entire pandemic is displayed in Figure 5. Clearly, in March–April 2020 there is a significant reduction in attendance to work and transit stations. Beyond this point, the annual fluctuations are fairly similar, indicating that people had adapted a “new normal” that they kept throughout the pandemic. Although we see recurrent significant oscillations, particularly due to summer and Christmas holidays, no significant fluctuations in behavior can be coupled with the rise and fall of new cases seen in Figure 1. Overall, this coincides with the authors personal observations, i.e., that while people updated their behavior during the pandemic, they did *not* continuously change social patterns depending on whether the viral levels were high or not.

However, in this case, the SEIR-type epidemiological models should in theory work, but no such model will predict herd-immunity at around 30% for a virus with the characteristics of SARS-CoV-2. Thus, we are left with two possible options to explain what is seen in Figure 3; either the models of SEIR-type are not useful, or some unknown factor was dampening the viral spread in society.

### 4.2. Hypothesis 2: Depletion of Susceptibles

We now update the model so that a(t), the human contact rate, is constant throughout the modeling time-span, but, instead, we include a certain level of artificial pre-immunity, which we choose manually in order to obtain a good fit with measured data. As discussed earlier, if variable susceptibility on an individual level exists, it will manifest itself as (artificial) pre-immunity on a macroscopic level [6]. Thus, if a certain level of pre-immunity, say 50%, gives a good fit between model and reality, it does not mean that 50% are immune to SARS-CoV-2. However, a certain level of protection against the Wuhan-strain may work less well against the alpha-strain, and, hence, there is no reason to assume that the level of artificial pre-immunity would be the same against the two strains (just as we assume that they have different transmission probabilities $\beta_{Wuhan}$ and $\beta_{alpha}$).

The model output, shown in Figure 6, uses a 65% artificial pre-immunity against the Wuhan strain and a 56% artificial pre-immunity against alpha, in addition to the 10% natural immunity acquired before September 2020. The blue, light brown, and yellow represent the measured incidence (total, Wuhan, alpha) just as in Figure 2. The green color shows the model-incidence due to the Wuhan strain and light blue represents the alpha strain, whereas dashed brown represent the total model-incidence. As is plain to see, the model gives a very accurate fit.

For the Wuhan-strain wave (green), we have $R_{e}\approx 1.4$ in early September 2020. To obtain a good fit for the subsequent alpha-wave, we adjust the artificial pre-immunity down from 65% to 56% and leave the transmission probability unchanged. This would mean that what made alpha dominant was not a superior spreading capacity among those already susceptible to the Wuhan strain, but rather that it mutated in a way that it could get around some of the pre-immunity. For the blue model curve (for alpha) we have an $R_{e}$ of around 1.5 in the initial phase. Note that this is perfectly compatible with reports of alpha being 38–130% more infectious, since the acquired immunity was much higher during the alpha wave, which means that the virus needs to be significantly more infectious in order to be able to keep spreading.

### 4.3. Herd-Immunity under NPIs

Under Hypothesis 2, the decline in cases seen in January and again in May 2021, is due to depletion of susceptibles, given the fixed and rather mild NPIs (or “recommendations”, as they were officially known). In Figure 6, we also plot the model output without the vaccination scheme (dashed pink curve), indicating that the cases caused by the alpha-strain would have started to decline in April anyway, indicating that some form of herd-immunity had been reached. Another indication of this is the fact that there was no notable surge among unvaccinated school children under the age of 12 (prior to the omicron-wave), as discussed in Section 2.4 and Section 4.1. Note that antibodies against alpha were shown to provide good protection against delta, but less so against omicron [26,27].

In any case, there is always the possibility that another surge could have followed if people had returned to pre-pandemic mobility patterns (which there is no indication of in Figure 5), so the term herd-immunity may be misleading. Therefore, we will refer to herd-immunity “under NPIs”. By this we mean the point in the model at which cases naturally start to drop due to depletion of susceptibles, leading to a value of $R_{e}$ below 1 (despite human social patterns assumed constant over time). How to accurately compute the point at which this happens (against a certain strain) is rather subtle, and described in detail in Section A.9.

If Hypothesis 2 is correct, then Stockholm reached herd-immunity under NPIs against the Wuhan-strain in early December 2020, just before the closing of high schools, when the incidence curve leveled out. This is because the herd immunity threshold is reached at the peak of the wave, not after it has receded, (which is why a major outbreak usually ends with a final size substantially larger than the herd-immunity threshold). By the same logic, the herd-immunity threshold under NPIs for the alpha-strain was reached on April 9, 2021, when most people in the group 0–69 were still unvaccinated.

## 5. Discussion

The SARS-CoV-2 hit-rate in Sweden prior to omicron was around 30%, about equally distributed over three distinct waves, in stark contrast with state of the art SEIR-models which predict at least 60% hit rate in one massive wave. It seems generally believed that this can be explained by NPIs and voluntary public reduction in contact patterns, as well as ultimately the vaccination campaign. We investigate this hypothesis and find that, based on data from serological studies, the incidence-curve and Google-mobility data, the hypothesis seems unlikely, even without doing any modeling. We then run an age-stratified SEIR model taking vaccination and isolation of elderly into account, and found that only an unrealistic decrease in contact patterns could explain the cases seen in Stockholm during chosen time frame.

This challenges the interpretation of variable NPIs and human behavioral changes as the *main cause* for the rise and fall of epidemic waves, at least in locations where authorities have been either unwilling or incapable of enforcing strict NPIs, such as Sweden. Although NPIs and changes in social behavior clearly are important, we demonstrate that it is difficult to envision that they alone can explain the outbreaks seen in Stockholm County during the second and third waves.

However, this raises the question: is it always so that SEIR-models significantly overestimated the magnitude of spread, or is there something particular about SARS-CoV-2 that kept the incidence unexpectedly low?

Although it is clear that people in Sweden made major changes to their social interaction patterns, it seems to us that these changes were kept rather constant prior to the vaccination campaign. Some people worked from home and minimized unnecessary contacts, others organized private parties, but in both cases they did so irrespective of whether the incidence was high or low. However, under the assumption that human behavior was rather constant, standard mathematical models, such as SEIR, should, in theory, work well to predict the spread of COVID-19 in a densely populated area, such as Stockholm County. However, we have showed that it clearly does not.

Another interesting case is that of Brazil, which also reported surprisingly low levels of seroprevalence from most major cities during the first year of the pandemic (possibly with the exception of Manaus) despite applying limited NPIs. Similar comments can be made also about India, which could be explained if a level of pre-immunity was indeed present against the major strains of SARS-CoV-2 prior to omicron.

To mathematically investigate this possibility, we rely on a SEIR-model which is an extension of a state of the art model (developed for modeling COVID-19 by well known specialists in the field [5]), and find that the model, under the assumption that everybody is equally susceptible, significantly overestimates the disease spread (recall Figure 3). The only way that we are able to obtain a good model fit with real data is by including a pre-immunity assumption, and we find that this pre-immunity fraction needs to be substantial, in the range 55–65%, in order to obtain a good match with measured data.

### 5.1. What Could Have Caused Pre-Immunity?

We demonstrate in the adjacent publication [6] that a number of factors, such as people isolating, variable social activity patterns, as well as variations in susceptibility, are manifested as “artificial pre-immunity”. Hence, the hypothesis of pre-immunity does not necessarily imply that certain people had sterilizing protection against SARS-CoV-2. However, 65% is a large proportion, and it seems far-fetched to assume that people isolating to various degrees could amount to an average effect of 65% artificial pre-immunity. Additionally, the fact that seroprevalence among unvaccinated school children leveled out at around 30% until the omicron-wave, seems to indicate that a majority of these children had a high level of protection against contracting the initial strains of SARS-CoV-2 (i.e., Wuhan, alpha, and delta).

### 5.2. The Case of Omicron

Relying on the numbers from the Swedish Public Health Agency for the unvaccinated group aged 0–11 (see Appendix A), we infer that the amount of antibodies in this group increased by 42% (from 29.6 to 71.8) during the omicron surge December 2021 to March 2022. Since it is known that antibodies against prior variants of SARS-CoV-2 gave a very limited protection against infection by omicron, this tells us that somewhere between 42.2% and 71.8% were infected with omicron in this wave, and based on the authors personal observations it seems that the higher number is the more accurate figure. The fact that around 40–70% were infected with omicron in the course of 3 months also gives support to the hypothesis that actually SEIR-models *do work*, but some other, yet to be fully understood, factor was damping the initial waves of SARS-CoV-2. Since the actual amount of people who were infected with omicron is unknown, we refrain from fitting the measured omicron-wave with our model, but whether it is around 40 or 70% it is clear that this can be completed without heavily relying on any form of pre-immunity (especially in the latter case).

If it is the case that antibodies due to some cross reactive immunity protected a large part of the population during the first waves, then natural selection will favor variants that are able to circumvent this protection. Indeed, the omicron-mutations managed to circumvent the antibodies from prior variants. Therefore, it is natural to postulate that also the level of pre-immunity gradually deteriorates and eventually disappears. Thus, the fact that there was a massive omicron-wave is not in contradiction with Hypothesis 2.

### 5.3. Influenza-A Mediated Pre-Immunity

A number of factors, such as genetic, cross-reactive immunity and innate immunity, have been shown to provide variation in susceptibility [22,23,24]. Due to the findings presented in this article, we searched for the identity of a protective pre-immunity to SARS-CoV-2 and found that previous infections by influenza A H1N1 (Flu–strains) provides a specific antibody-mediated immunity preventing binding of the Spike protein to the Angiotensin Converting Enzyme 2 (ACE2) receptor. This antibody reactivity was found in 55–73% of people in Stockholm [7] and could, hence, be an important factor behind the 65% “artificial” pre-immunity identified in the present study. We also identified 12 different cross reactive T cell peptides that are shared between Influenza A H1N1 and SARS-CoV-2, which could provide HLA dependent T cell mediated protection against severe COVID-19 disease. We provide modeling data that Scandinavians carry HLA types that could mediate such protection in 71% of the population, while only in 40% of people among the world’s population [7].

We highlight that the fact that the hit-rate was rather low in Sweden, and that the death rates were lower in Sweden than in some other countries, fits well with this theory. In our study describing the identity of the Flu mediated protective pre- immunity, we make a case analysis with Stockholm and India to support this statement. Yet more compelling evidence that this theory could be true in reality comes from epidemiological observations demonstrating that Flu vaccinations have protected people from becoming infected with SARS-CoV-2 or becoming severely ill or dying from COVID-19 [28,29,30,31,32,33], with estimated protection rates of about 40–80%.

When omicron emerged, we were concerned that the mutations in the receptor binding domain of the Spike-protein (in region 477–505) would cause a loss of immunity mediated by previous infections, vaccinations, as well as the Flu mediated cross-protective immunity (localized to region 481–486) and create a highly infectious variant. As a consequence, we expected that many people would become infected with omicron, which is precisely what then happened. In this scenario, the observed data in Sweden and elsewhere is completely compatible with SEIR-models, and the fact that the outcome of the Swedish relaxed strategy was not a complete disaster could be attributed to luck, as this protective immunity was not known to the Swedish authorities. This possibility is important to be aware of when modeling infectious disease outbreaks and forming strategies to protect people in future pandemics.

## 6. Conclusions

We find that state of the art mathematical models, as well as standard formulas for the herd immunity threshold, are completely at odds with what happened in Sweden during the first year of the pandemic. The hypothesis that this is due to non-pharmacological interventions and/or voluntary updates in social patterns, is found unlikely to be true.

We are, thus, left with two possibilities: either state of the art models are inapt for modeling COVID-19, or a large proportion of the population had some a priori protection against contracting SARS-CoV-2. This is as far as the present work takes us, and we can only speculate over which answer is correct. Pre-immunity is one possible explanation for why there is such a large gap between model output and observed data, and, in this case, data are completely compatible with SEIR-models.

In summary, we argue that our findings favors the hypothesis that state of the art SEIR-models are good for modeling spread of SARS-CoV-2, but that a large fraction of the population indeed had some degree of protective pre-immunity, due to factors that remain to be understood in greater depth.

## Figures and Tables

**Figure 1 viruses-14-01840-f001:**
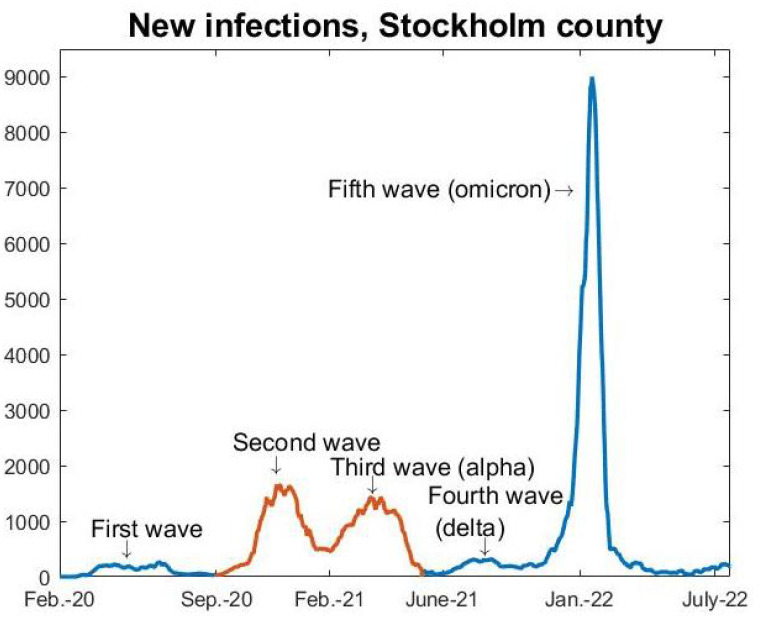
Full data series (weekly average) from the Swedish Public Health Agency. During first wave, only patients admitted to hospital were tested, so the data are very unreliable. General testing started in late May 2020. The red section indicates the time frame we focus on in this paper.

**Figure 2 viruses-14-01840-f002:**
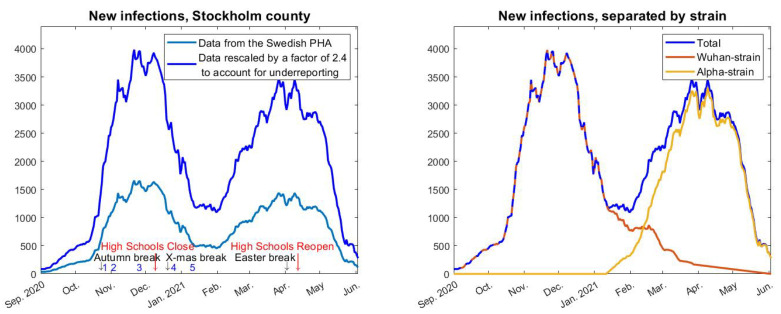
(**Left**): 7-day rolling average of new cases in Stockholm county, adjusted to match the reported increase in seroprevalence of antibodies against SARS-CoV-2. The numbers 1–5 correspond with the updates to the NPIs in Table 1. (**Right**): total cases separated in Wuhan-strain and Alpha-strain.

**Figure 3 viruses-14-01840-f003:**
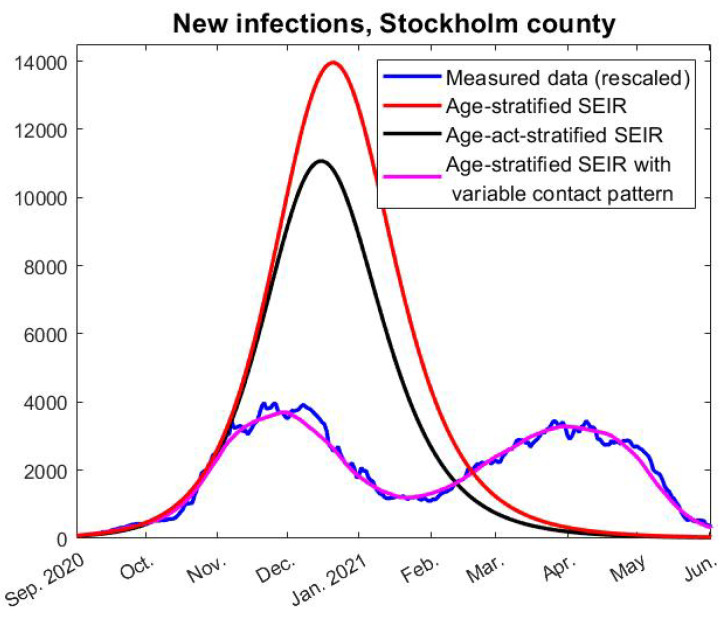
Comparison between real data and model outcome for the age-stratified SEIR and age-activity-stratified SEIR based on [5], which assumes that contact patterns are stable over time. The pink graph is produced with variable social interactions (over time), as described further in Section 4.1.

**Figure 4 viruses-14-01840-f004:**
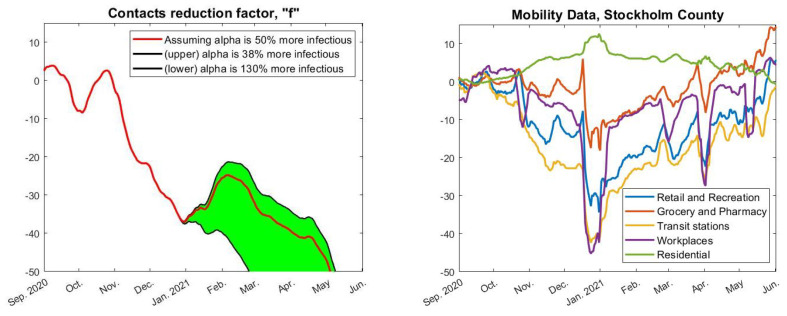
(**Left**): relative change in daily contacts needed to obtain a good fit with real data. The green area accounts for uncertainty about how much more contagious the alpha-strain was. The red central curve assumes 50% more contagious (following [16]), and the (almost perfect) data fit in Figure 3 (pink curve) is computed based on this. (**Right**): actual reduction in mobility pattern according to Google Mobility Data. Notice that the y-axis is the same in both the left and the right image, and that one would expect to see similar curves given that Hypothesis 1 holds.

**Figure 5 viruses-14-01840-f005:**
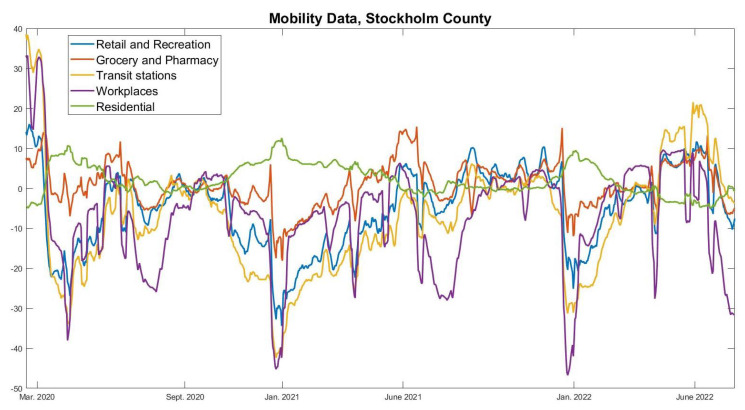
Google-mobility data over the full pandemic time-frame, preprocessed as in Figure 4.

**Figure 6 viruses-14-01840-f006:**
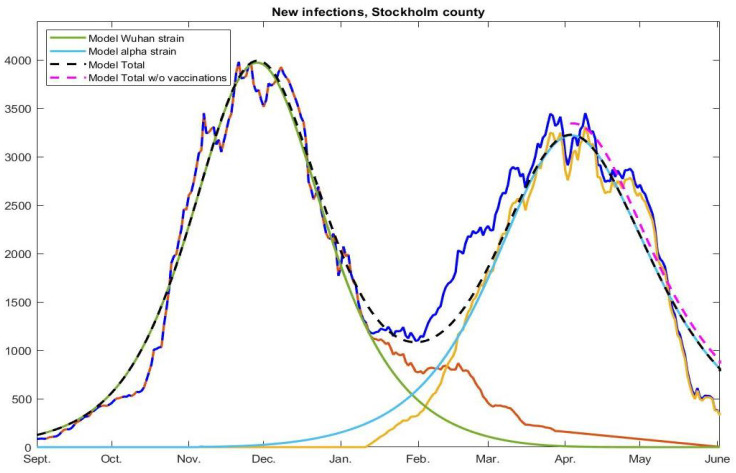
Actual and modeled curves using artificial pre-immunity. The brown-orange, yellow, and blue curves display measured cases, separated into Wuhan-strain, alpha-strain, and total, respectively, same as in Figure 2 (so reweighted as before). The green, light-blue, and dashed black display corresponding curves for our model. Finally, the pink dashed curve shows total model output in the absence of vaccinations (only displayed during April–May 2021). Clearly, if this model is correct, the vaccination campaign had virtually no part in the final demise during May 2021, contrary to popular belief.

**Table 1 viruses-14-01840-t001:** Summary of major updates to the NPIs in Sweden between September 2020 and May 2021.

1 (Oct-20)	Recommendations for elderly (70+) to self-isolate are removed and replaced by the same recommendations as for the general population.
2 (Nov-20)	Before 1 November the maximum amount of visitors to a public event was 50, which then was changed to 300, although a cap on maximum 50 participants in dancing events remained.
3 (Nov-20)	Prohibited public gatherings involving more than 8 people (shopping, restaurants, bars etc., were exempt from this rule).
4 (Dec-20)	Alcohol-sales only until 20.00, compared with 22.00 previously.
5 (Jan-21)	Facemask recommendation is issued for health care visits and rush-hour public transport.

## Data Availability

Codes and data are available on https://github.com/Marcus-Carlsson/Covid-modeling (accessed on 10 Augsut 2022) as well as in Appendix A.

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
