# Peer review of "COVID-19 Modeling Outcome versus Reality in Sweden"

_viruses, 2022, doi:10.3390/v14081840_

Round 1

Reviewer 1 Report

The authors present an analysis of a SEIR (with age-compartments) model where in particular it is shown that the inclusion of some pre-immunity (unknown mechanism) can numerically describe COVID-19 data from the city of Stockholm, Sweden.

The study is restricted to the so-called second- and third-waves of high activity in Sweden, and it is well argued in the paper why this restriction in time is chosen. It should be noted that just the fact that two waves are modelled makes it alone complicated compared to a standard SEIR which only gives one wave.

One may argue that many details of society is missing and/or left without a medicine-based explanation in the paper. For example, interaction between age groups data is taken from an older Dutch study, though adjusted to suitable Swedish age groups (70+).
However, in my view, it is an important strength of the paper that the authors do not focus on too many detailed explanations or estimations of many parameters. Instead, they show that with a low complexity model, explained well in the paper, and with ‘standard’ parameter values, they can obtain agreement with data from Stockholm just by introducing additional pre-immunity.

They give suggestions for what the (artificial) pre-immunity can be, but leave it for future work to examine some of the candidate explanations in detail.
Unless a medicine-based explanation can be found, it may be hard to be sure about the actual mechanism.
For example, a large part of the citizens starting to follow a “2m distance” recommendation may have effect, but may not be captured by “mobility data”.

In relation to the important large scale and high costl decisions for countries and regions taken by politician, backed up by scientist to different degrees in different countries, during the early years of the pandemic. I think that this kind of research is very important as helping to build a base of useful knowledge for the next pandemic.

Below follows some minor comments that can help to improve the presentation:

The two “hypothesis 1) and 2)” are clearly stated in the middle of page 2. However, a few lines above that, they are referred to as “above”?

Text/titles and labels in Figs. 1, 2, A1 should be written with a larger font.
Fig. A1, please also state the year in the caption.

Since figures and tables should preferable be readable without referring to the text, indicate what years (and country) Table 1 is relevant for in the caption.

Please improve the explanation for some of the abbreviations used, for example:
*PHA is now explained on page 4, but appears earlier.
*SEI -groups, explain.
*ICU, explain.

I later discovered that some of those mentioned above was explained in an “Abbreviations” section in the end of the paper. But for readability I recommend to explain theme as they appear in the text (or in the beginning of the paper).

In Fig. 3 several different colours (and linestyles) are used. Some explained in the legend, some in the caption, some only in the text. While I found this slightly confusing I was able to decode it, but there is room for improvement. For example the curve referred to as “light brown” in the text, do not appear to be brown to me.

Inconsequent writing: “seroprevalence” or “sero-prevalence”?

There is a “the” too much on lines 539 and 560 (top).

Refs. [11] and [12], the Swedish letter ä is missing the dots.

Ref. [14], use capital ‘L’ in “The lancet”.

Author Response

Comment: The authors present an analysis of a SEIR (with age-compartments) model where in particular it is shown that the inclusion of some pre-immunity (unknown mechanism) can numerically describe COVID-19 data from the city of Stockholm, Sweden.

The study is restricted to the so-called second- and third-waves of high activity in Sweden, and it is well argued in the paper why this restriction in time is chosen. It should be noted that just the fact that two waves are modelled makes it alone complicated compared to a standard SEIR which only gives one wave.

One may argue that many details of society is missing and/or left without a medicine-based explanation in the paper.

Answer: This is true but we remark that this is standard in epidemiological models and we make a case that the magnitude of the output of our model does not deviate from modeling teams that have published their works in various top journals. Indeed, one of the key points we establish is precisely that standard models are insufficient to describe what happened in Sweden, and therefore it is important to use standard models rather than some homecooked advanced code. This alone is an interesting observation that does not seem to be acknowledged or discussed in the publications we have seen from other modeling teams.

For example, interaction between age groups data is taken from an older Dutch study, though adjusted to suitable Swedish age groups (70+).

Answer: The contact matrix is also calibrated, and we checked that the distribution of the viral spread in the various groups actually coincides well with measured data by the Swedish Public Health Agency, indicating that the matrix is a good reflection of the actual society. Moreover, the model is pretty robust and changes in A does not have any major effect on the overall behavior.

However, in my view, it is an important strength of the paper that the authors do not focus on too many detailed explanations or estimations of many parameters. Instead, they show that with a low complexity model, explained well in the paper, and with ‘standard’ parameter values, they can obtain agreement with data from Stockholm just by introducing additional pre-immunity.

They give suggestions for what the (artificial) pre-immunity can be, but leave it for future work to examine some of the candidate explanations in detail.
Unless a medicine-based explanation can be found, it may be hard to be sure about the actual mechanism.

Answer: This is true, but we discuss this topic in further detail in the revised version, Section 5.

For example, a large part of the citizens starting to follow a “2m distance” recommendation may have effect, but may not be captured by “mobility data”.

In relation to the important large scale and high costl decisions for countries and regions taken by politician, backed up by scientist to different degrees in different countries, during the early years of the pandemic. I think that this kind of research is very important as helping to build a base of useful knowledge for the next pandemic.

Answer: Thank you, we agree and we have now added a graph, Figure 3, which clearly demonstrates how much standard models tend to overshoot, compared to what was measured in Sweden.

Below follows some minor comments that can help to improve the presentation:

The two “hypothesis 1) and 2)” are clearly stated in the middle of page 2. However, a few lines above that, they are referred to as “above”?

Answer: We have rephrased this entire section and hope it is clearer now for the readers.

Text/titles and labels in Figs. 1, 2, A1 should be written with a larger font.
Fig. A1, please also state the year in the caption.

Since figures and tables should preferable be readable without referring to the text, indicate what years (and country) Table 1 is relevant for in the caption.

Answer: We thank the reviewer for these suggestions, which we have amended the text accordingly.

Please improve the explanation for some of the abbreviations used, for example:
*PHA is now explained on page 4, but appears earlier.
*SEI -groups, explain.
*ICU, explain.

I later discovered that some of those mentioned above was explained in an “Abbreviations” section in the end of the paper. But for readability I recommend to explain theme as they appear in the text (or in the beginning of the paper).

Answer: Thank you for pointing this out, we removed the abbreviation PHA altogether and have tried to improved readability.

In Fig. 3 several different colours (and linestyles) are used. Some explained in the legend, some in the caption, some only in the text. While I found this slightly confusing I was able to decode it, but there is room for improvement. For example the curve referred to as “light brown” in the text, do not appear to be brown to me.

Answer: We thank the reviewer for pointing this out, which is key for understandability of the overall reasoning. We have tried to improve the figure text legends in all figures. Concerning light-brown, we now wrote brown-orange… we hope this is a better choice.

Inconsequent writing: “seroprevalence” or “sero-prevalence”?

There is a “the” too much on lines 539 and 560 (top).

Refs. [11] and [12], the Swedish letter ä is missing the dots.

Ref. [14], use capital ‘L’ in “The lancet”.

Answer: Thank you, we have amended the text accordingly.

Reviewer 2 Report

On December 2019, a cluster of pneumonia cases of unknown etiology was reported in Wuhan (China). The cases were declared to be Coronavirus Disease 2019 (COVID-19) by the World Health Organization (WHO).  Sweden introduced a more permissive policy (no or low restrictions) during the period studied in this paper. Therefore, it is a very interesting case study for evaluating COVID-19 trend.

The introduction is well written, otherwise related works are poor. The introduction could be divided into Introduction and Background. Furthermore, authors mentioned the Swedish Health Agency (line 45) as data source, in my opinion you could add a reference to the dataset, e.g.: other works, official web-site, or something that allows reader to access this one.

-        Sweden cases related to Covid-19 have been analyzed and compared to other countries (i.e., Italy, France, UK, USA) in the following paper: doi.org/10.3390/e24070929  I suggest including it within the references, in that it highlights the relevance of the policies adopted by countries in the last years.

"Methods" is very exhaustive when the model is explained, but it needs to be improved, explaining also how the model is applied. To give some example: (i) Has the model been implemented? (ii) What environment was used to develop and test the model? (iii) Have you made use of technologies (e.g., software tools, or framework) already available in the literature? Some of these questions are answered in the Appendix A.1, however, in my opinion, the authors should integrate this information in “Methods”.

The results are well presented and discussed. However, I think that a statistical test of significance could be useful to corroborate the results.

Conclusion are missing. In my opinion, a section for conclusion could be useful to summarize the key points of the manuscript, reporting also the essential information related to the results obtained during the test and discussed in “Discussion”. The authors could explain very briefly what has been done and what has been demonstrated, confirming the importance of the article.

Minors:

-        I suggest checking sections’ numbering, for instance the Introduction is reported as 0, I think its number should be 1

-        Spell checking and typos.

Author Response

On December 2019, a cluster of pneumonia cases of unknown etiology was reported in Wuhan (China). The cases were declared to be Coronavirus Disease 2019 (COVID-19) by the World Health Organization (WHO).  Sweden introduced a more permissive policy (no or low restrictions) during the period studied in this paper. Therefore, it is a very interesting case study for evaluating COVID-19 trend.

The introduction is well written, otherwise related works are poor. The introduction could be divided into Introduction and Background. Furthermore, authors mentioned the Swedish Health Agency (line 45) as data source, in my opinion you could add a reference to the dataset, e.g.: other works, official web-site, or something that allows reader to access this one.

-        Sweden cases related to Covid-19 have been analyzed and compared to other countries (i.e., Italy, France, UK, USA) in the following paper: doi.org/10.3390/e24070929  I suggest including it within the references, in that it highlights the relevance of the policies adopted by countries in the last years.

Answer: We separated the introduction in background and contributions. Also, the accessibility of data is very important so we have created an Appendix B that explains how to retrieve all data, along with a GitHub location where our code can be downloaded. We also included the suggested reference, thank you for this proposal.

"Methods" is very exhaustive when the model is explained, but it needs to be improved, explaining also how the model is applied. To give some example: (i) Has the model been implemented? (ii) What environment was used to develop and test the model? (iii) Have you made use of technologies (e.g., software tools, or framework) already available in the literature? Some of these questions are answered in the Appendix A.1, however, in my opinion, the authors should integrate this information in “Methods”.

Answer: We included a few more details in Section 3.1 as well as uploaded all relevant codes and data to GitHub. Also, as a sort of stability test vis a vis different models, we also implemented the Age-Activity stratified SEIR by Britton et al and display corresponding graph in Figure 3. We added a few more remarks on how the code was made, but refrain from discussing implementational details too profoundly since it has no bearing on the graphs. To confirm this point, we actually have implemented the code both in Python and MatLab, using both the standard ODE solver ode45 as well as a simple for loop with a daily step count.

The results are well presented and discussed. However, I think that a statistical test of significance could be useful to corroborate the results.

Answer: Thank you for this suggestion. However, while performing a significance test is a good idea in principle, this is difficult to do with the present material, hence we have not followed up on this suggestion.

Conclusion are missing. In my opinion, a section for conclusion could be useful to summarize the key points of the manuscript, reporting also the essential information related to the results obtained during the test and discussed in “Discussion”. The authors could explain very briefly what has been done and what has been demonstrated, confirming the importance of the article.

Answer: Thank you for this suggestion, we have included a new section with key conclusions of this work.

Minors:

-        I suggest checking sections’ numbering, for instance the Introduction is reported as 0, I think its number should be 1

-        Spell checking and typos.

Section numbering has been checked.

We have thoroughly performed spell checking for typos.

Round 2

Reviewer 2 Report

The paper has been revised well; so I recommend it for publication.